# Exploring the Quality of Life Related to Health and Vision in a Group of Patients with Diabetic Retinopathy

**DOI:** 10.3390/healthcare10010142

**Published:** 2022-01-12

**Authors:** Ian Roberts-Martínez Aguirre, Paula Rodríguez-Fernández, Josefa González-Santos, Nerea Aguirre-Juaristi, Nuria Alonso-Santander, Juan Mielgo-Ayuso, Jerónimo J. González-Bernal

**Affiliations:** 1Hospital of Mendaro, Integrated Health Organization Debabarrena, 20850 Gipuzkoa, Spain; i.roberts.ma@googlemail.com; 2Department of Health Sciences, University of Burgos, 09001 Burgos, Spain; nasantander@ubu.es (N.A.-S.); jfmielgo@ubu.es (J.M.-A.); jejavier@ubu.es (J.J.G.-B.); 3Point of Continued Attention Iztieta, Integrated Health Organization Donostialdea, 20004 Gipuzkoa, Spain; nereagi87@gmail.com

**Keywords:** diabetes mellitus, diabetic retinopathy, visual function, care quality improvement, quality of life

## Abstract

(1) Background: Visual impairment of people with diabetic retinopathy (DR) and its high impact on different dimensions of their lives can cause a significant deterioration in the quality of life. The aim of this study was to examine the association and relationship between quality of life related to vision and the relevant clinical and sociodemographic variables in a group of patients with DR in Spain. (2) Methods: A descriptive cross-sectional study was conducted in all patients with DR over 18 years under follow-up in the Retina Service of the University Hospital of Burgos (HUBU), recruited during the months of January and February 2020. The main study variable was quality of life related to health and vision, obtained using the National Eye Institute Visual Function Questionnaire 25 (NEI-VFQ-25). (3) Results: In total 87 participants made up the sample, and significant differences were found in the NEI-VFQ-25 according to gender, type of diabetes, episodes of decompensated diabetes and high blood pressure (HBP) (*p* < 0.05). Best-corrected visual acuity (BCVA) was also correlated with the NEI-VFQ-25 (*p* < 0.05). (4) Conclusions: These data could facilitate the design of action protocols focused on the well-being of the patient, in addition to considering the clinical characteristics. Further studies are needed to help understand the causal relationship between variables and that includes a wider variety of factors.

## 1. Introduction

Diabetes mellitus (DM), one of the most important public health challenges of the 21st century, is frequently considered a global epidemic [1]. It is an increasingly frequent pathology among the urban population of developed and developing countries and will increase significantly during the 2020s [2,3]. Recent European studies, based on health service records, report DM incidences of between 3 and 6 cases per 1000 person-years [2,3], and a prevalence of 4.4% is expected in the year 2030 and an increase in the total number of diabetics from 171 million in 2000 to 366 million in 2030 [4]. Based on current data from global studies, the International Diabetes Federation (IDF) estimates that the number of people with DM aged 18 to 99 years globally will reach 693 million by 2045 [5].

Diabetic retinopathy (DR) is one of the microvascular complications of long-term, poorly controlled DM [6]. It is a progressive disease that causes deterioration and loss of vision in the population; fluctuating, blurred, double or distorted vision; floats in the field of vision; or changes in refractive error [6]. 

According to the World Health Organization (WHO), DR is responsible for 5% of blindness worldwide, with percentages that are up to 15–17% in some countries [7], and being the main cause of preventable blindness in adults under 75 years of age in developed countries [8,9]. Between 12,000 and 24,000 people with DM develop visual impairment each year, which represents 12% of new annual cases of blindness [7]. Almost all people with type 1 diabetes (T1D) and more than 60% of patients with type 2 diabetes (T2D) are diagnosed with DR 15 to 20 years after the onset of DM [10]. In Spain, there has been a great variety in prevalence studies since the year 2000, where the proportion of people suffering from the disease ranges between 7.20% and 37.50% due to the methodological variability of the investigations [11].

Vision plays an important role in information processing and in the interaction with the environment, as well as in the performance of daily activities [12,13]. Visual impairment [14], and more specifically suffering from RD, has been related to dependence on activities of daily living [15,16], social isolation [17], and reduced physical activity [18]. In this line, the visual impairment of people with DR and its high impact on the different dimensions of their lives can cause a significant deterioration in the quality of life [19,20,21,22].

It is important to note that DR has been shown to be a threat to the quality of life both in patients with T1D and T2D [23]. DM usually presents late complications or comorbidities that can be significantly associated with the quality of life of people with DR, beyond visual impairment and its consequences [24,25]. In this line, the quality of life of people with DR is significantly lower than that of people without this pathology [19,20,21,22], so considering this aspect could help decision-making in relation to the type of treatment and the moment of its implementation, as well as to monitor the patient’s responses to the treatment. There are many methods to measure the quality of life in specific diseases, and one of the most commonly used questionnaires to assess the vision-related quality of life is the National Eye Institute Visual Function Questionnaire (NEI-VFQ-25) [26].

Studying the quality of life of people with DR could offer valuable information on how the disease affects all aspects of their health and has become the cornerstone for the provision of comprehensive care oriented to the particular needs of each person. Therefore, the objective of this study was to study the association and relationship between the quality of life related to vision and relevant clinical and sociodemographic variables in a group of patients with DR in Spain.

## 2. Materials and Methods

### 2.1. Study Design—Participants

A descriptive cross-sectional study was conducted. All patients under follow-up in the Retina Service of the University Hospital of Burgos (HUBU), with a diagnosis of DR according to the ETDRS classification [27], and age equal to or greater than 18 years were included in this study.

Subjects with an incomplete medical history were excluded.

### 2.2. Procedure

Participants were recruited during January and February 2020 through convenience sampling. The doctor from the Retina service collected the data. The possible candidates were previously informed of the purpose and procedure of the study during the consultation, and if they wanted to participate an informed consent form was required. Data on the quality of life related to health, vision and sociodemographic variables were obtained through a self-administered questionnaire. Once all the study variables had been collected, statistical analysis of the data was performed and the results were interpreted.

The study received a favorable report from the Ethics Committee for Research with Medicines (Royal Decree 1090/2015 on Clinical Trials with Medicines) and was carried out in accordance with the ethical principles of the Declaration of Helsinki.

### 2.3. Main Outcomes—Instruments

The main study variable was quality of life related to health and vision. It was obtained using the NEI-VFQ-25 [26], a self-administered vision-specific patient-reported outcome measure, reporting on visual function in everyday life, which has been validated worldwide across different ocular diseases. NEI-VFQ 25 has also been considered a tool that is sensitive to change in visual acuity as it measures general health, general vision, eye pain, difficulty in near vision, difficulty in distant vision activities, limitation in social functions due to vision, mental problems due to vision, role problems due to vision, dependence on other people, driving difficulties, color vision problems and peripheral vision difficulties. The total score can range from 0 to 100; higher scores indicate better results [26].

The NEI-VFQ-25 is adapted for the Spanish population, with adequate validity and reliability α = 0.831 (95% CI: 0.735–0.904) in the entire questionnaire and α > 0.70 in all subscales except in “driving”, which, as in its official version, obtained lower reliability because it is an activity that not everyone performs, thus reducing the response rate [28]. The approximate time dedicated to completing the questionnaire was 10 min.

Data from the medical history were also collected, including gender (female/male); age; type of diabetes (T1D/T2D); insulin treatment (yes/no); episodes of diabetic decompensation (yes/no), understood as hospitalization for hypoglycemia, or hyperosmolar coma during the last year; nephropathy (yes/no); HBP (yes/no); time since diagnosis of diabetes and since diagnosis of DR; and glycosylated hemoglobin (HbA1c).

An eye examination was also performed to measure the best-corrected visual acuity (BCVA). First, the spontaneous vision of each eye was independently taken with a decimal scale test. If visual acuity was under 1.00 in any eye, the correction obtained with an autorefractometer was applied in trial frames, and the vision of each eye was taken again independently. Once both eyes have been scored, the best visual acuity obtained is considered as the BCVA. The score ranges from 0.05 to 1, with higher scores reflecting better functioning of the visual system. Finger counting, hand motion and light perception were considered in patients with underscale visual acuity.

### 2.4. Statistical Analysis

Descriptive analyses of the characteristics of the sample were carried out, and categorical variables were expressed in terms of absolute frequencies and percentages, and continuous variables as means and standard deviations (SD). The normality of the data set was contrasted using the Kolmogorov–Smirnov test. To evaluate the association between the quality of life related to health and vision and the categorical variables, the Mann–Whitney test was used. Spearman’s correlation was used to assess the relationship between quality of life related to health and vision and different continuous variables.

Statistical analysis was performed with SPSS version 25 software (IBM Inc., Chicago, IL, USA). For the analysis of statistical significance, a value of *p* < 0.05 was established.

## 3. Results

The present study consisted of 87 subjects with the diagnosis of DR. Details such as gender, type of diabetes, diabetic complications, and comorbidities are included in Table 1.

Among the participants, the male gender predominated (*n* = 60), and the mean age ± SD was 67.57 ± 10.88 years. A large majority of study subjects (82.8%) had T2D and 67.8% (*n* = 59) had insulin treatment. The participants had a mean ± SD of 19.45 ± 9.91 years with the diagnosis of DM and 6.39 ± 4.73 years with DR (Figure 1). The mean HbA1c level at the time of NEI-CFV-25 among the cases was 7.42 mg/dl (SD ± 1.57), and a BCVA of 0.716 (SD ± 0.25).

The mean score ± SD of NEI-VFQ-25 was 72.24 ± 6.21, the minimum score obtained being 54 points and the maximum being 94 points (Figure 2).

When comparing the ranges of the groups that make up the categorical variables of the study, statistically significant differences were found based on gender in the subscale dependence, driving, color vision, and NEI-VFQ-25 total, the female gender being the one that reported the worst results. In those related to diabetic characteristics and complications, participants with T1D demonstrated poorer mental health than those with T2D, and greater dependence, worse color vision and peripheral, and a poorer quality of life related to vision in general in subjects who had been admitted for diabetic decompensations. No statistically significant differences were found between insulin treatment and non-insulin treatment. Neither were significant differences found between groups in relation to comorbid pathologies, with the exception of HBP, where it was found that people with this pathology were significantly more dependent than participants who did not suffer from it (Table 2).

None of the continuous variables were correlated with the NEI-VFQ-25 scale, except for the best-corrected visual acuity. As visual acuity increased, the subscale scores for general vision, close activities, distance activities, and mental health decreased, and the scores of the subscales of role difficulties, dependence, driving ability, and color vision increased. Therefore, people with higher corrected visual acuity perceived poorer vision in general, poorer performance of near and far activities, and poorer mental health, but they also reported fewer role difficulties, less dependency, fewer driving difficulties, and better color vision (Table 3).

## 4. Discussion

The great advances in medical care suggest advantages of more patient-centered approaches, considering aspects such as well-being or quality of life to be especially relevant to tailor treatments. In the case of DR, one of the main causes of blindness in developed countries, numerous studies have shown a significant reduction in the quality of life of patients with DR compared to those without DR. Knowing the factors associated with this decrease could facilitate the design of action protocols focused not only on reducing or eliminating the symptoms of DR, but also on improving the quality of life and well-being of people suffering from the disease.

In the present study, the NEI-VFQ-25 was used to evaluate the quality of life related to health and vision, and a score of 72.24 ± 6.21 was obtained. A recent study evaluated the quality of life of people with RD and that of people without RD using the same evaluation instrument and showed statistically significant differences between both groups, with a score of 73.93 ± 25.55 in the group of people with RD and 99.26 ± 1.01 in the group of people without RD [1]. Taking these data into account, a decrease in the quality of life of people with DR in this study can be affirmed. This result agrees with many studies that have also demonstrated a qualitative and quantitative decrease in quality of life in this group [1,7,10,12,29,30,31].

The objective of this research was to study the quality of life related to health in a group of patients with the diagnosis of DR in Spain, taking into account sociodemographic and clinical data; statistically significant difference in the quality of life was observed between groups in variables such as gender, the type of diabetes, suffering episodes of diabetic decompensation or hypertension, and BCVA.

Women with DR reported a poorer quality of life in general compared to the male gender and also showed greater dependency, poorer driving ability, and poorer color vision than men. Trento et al. found in their study, to evaluate changes in quality of life related to vision in patients with DR and vision problems using the NEI-VFQ-25, worse driving in women with DR and higher scores on the subscales of general vision, close activities, distance activities, specific visual social functioning, mental health, and color vision [12].

People with T1D showed poorer scores on the mental health subscale compared to those with T2D, and although most of the existing studies focus on children or adolescents due to the characteristics of T1D, previous research has shown a significant association of this type of diabetes with a poorer quality of life [32]. Glycemic control has been widely studied in diabetes clinical research [33], with HbA1c levels and episodes of diabetic decompensation being the ones who reported a higher risk of having a poorer quality of life related to health. All these results partially coincide with those obtained in the present investigation, because participants with DR with episodes of diabetic decompensation and worse mental health reported lower health-related quality of life scores, but HbA1c levels were not correlated with quality of life. The mean HbA1c level was 7.42 mg/dl (SD ± 1.57), and the goal of HbA1c in the treatment of diabetic patients is around an HbA1c level ≤7% [34], which suggests patients with fairly controlled diabetes in this study. Pereira et al. [1] also did not find a correlation between NEI-VFQ-25 scores and HbA1c levels, but they did find a significant relationship between the duration of diabetes and the study variable. Furthermore, the duration of DR and comorbidities also affected the quality of life of patients with DR. These results coincide with the findings of Alcubierre et al., who demonstrated that the quality of life is affected by the severity and duration of DR, as well as by insulin therapy [7].

In this study, no significant correlations were found between quality of life and being insulin-dependent or having comorbid diseases, with the exception of hypertension, which was revealed to promote greater dependence. The duration of both DM and DR were also not correlated with the health-related quality of life and vision of patients with DR in this study.

The main aspect related to the study variable was BCVA. A higher BCVA was correlated with reduced scores in general vision, close activities, distance activities, and mental health subscales. It was also correlated with high scores in the difficulty of role, dependence, driving ability, and color vision. These results partially coincide with those of Trento et al., whose multivariate analysis showed that reduced visual acuity was associated with reduced scores for general vision, close activities, distance activities, social functioning, health mental, role difficulties, driving, color vision, and peripheral vision [12]. DR and vision loss alter people’s perception of functional capacity [12]. The fact that people with a higher score on BCVA sometimes report worse results in the different dimensions of quality of life related to health and vision may be due to the fact that, as it is a chronic disease, patients end up reconciling and getting used to their condition and learn to live with impaired vision [20].

This study provides information about the quality of life related to the health and vision of people with DR, which facilitates a patient-centered approach and aids in clinical decision-making concerning treatment. It is worth mentioning some limitations of our study, such as that a larger sample size could be more representative of the population and increase the precision of the new estimated parameters. It is important to mention that data collection was interrupted by the global pandemic derived from the disease caused by the SARS-CoV-2 virus (COVID-19), and data could only be obtained during the months of January and February 2020. In addition, obtaining the sample from a single center also reduces the representativeness of the results, so convenience sampling has also been able to induce methodological biases in the study. Another limitation is that, despite the fact that a series of variables related to diabetes could be accessed, it is likely that other relevant comorbidities and sociodemographic variables were not considered, which may have influenced the results. Moreover, the cross-sectional design did not allow to study the causality. Further longitudinal studies are needed to understand the causal underpinnings of quality of life related to the health and vision in people with DR, and to examine possible predictive factors. In addition, it would be convenient to collect and analyze a wide set of variables that includes socioeconomic, emotional, and lifestyle-related aspects of the population studied.

## 5. Conclusions

The quality of life related to health and vision of people with DR differs depending on factors such as gender, type of diabetes, suffering episodes of diabetic decompensation, hypertension, and BCVA. These data could facilitate the design of action protocols focused on the well-being of the patient, in addition to considering the clinical characteristics. Further studies are needed to help understand the causal relationship between variables and that include a wider variety of factors.

## Figures and Tables

**Figure 1 healthcare-10-00142-f001:**
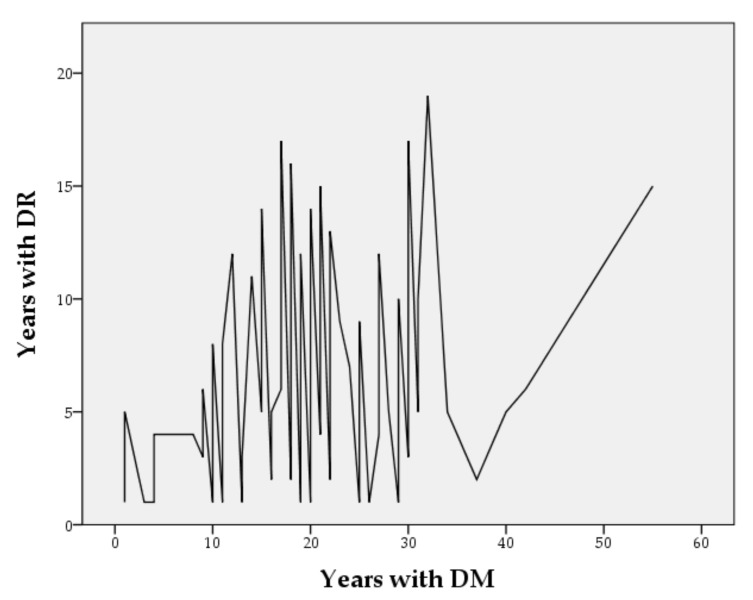
Years since the diagnosis of DM and the moment of appearance of DR in the study sample.

**Figure 2 healthcare-10-00142-f002:**
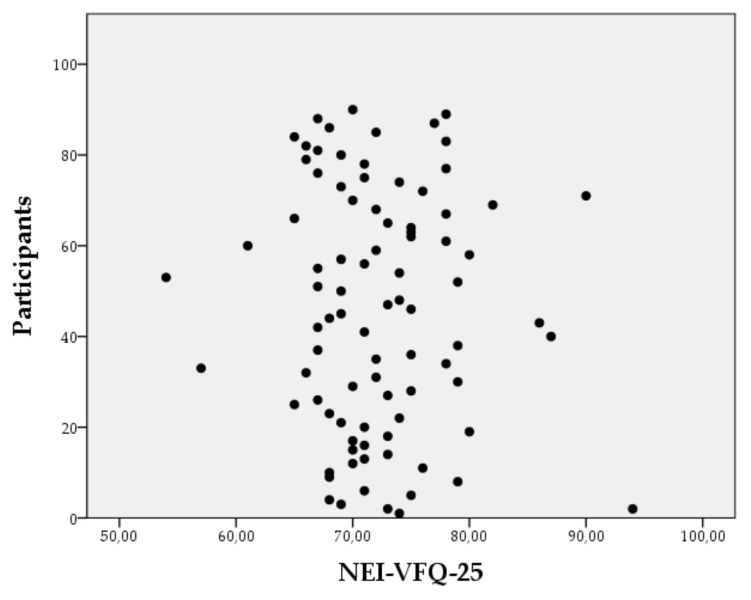
Distribution of the participants’ total score on the NEI-VFQ-25.

**Table 1 healthcare-10-00142-t001:** Characteristics of the sample.

Categorical Variables		*n*	Percentage
Gender	Male	60	69%
	Female	27	31%
Diabetes type	T1D	15	17.2%
	T2D	72	82.8%
Insulin treatment	Yes	59	67.8%
	No	28	32.2%
Diabetic decompensation	Yes	16	18.4%
	No	71	81.6%
Nephropathy	Yes	19	21.8%
	No	68	78.2%
High Blood Pressure	Yes	65	74.7%
	No	22	25.3%
Hypercholesterolemia	Yes	60	69%
	No	27	31%

Arterial hypertension; *n*: number of participants.

**Table 2 healthcare-10-00142-t002:** Analysis of the mean ranges of the subscales and total score of the NEI-VFQ-25 according to the categorical variables.

NEI-VFQ-25	Gender		Diabetes Type		Insulin Treatment		Decompensated Diabetes		Nephropathy		AHT		Hypercholes Terolemia	
	M	F	1	2	Yes	No	Yes	No	Yes	No	Yes	No	Yes	No
General health	42.41	47.54	46.93	43.39	45.16	41.55	46.38	43.46	52.47	41.63	46.28	37.27	45.79	40.02
General vision	42.33	47.72	37.50	45.35	42.53	47.09	43.06	44.21	43.68	44.09	45.52	39.50	45.24	41.24
Ocular pain	43.16	45.87	45.63	43.66	44.11	43.77	42.78	44.27	40.61	44.95	44.25	43.27	44.54	42.80
Near activities	42.33	47.70	37.63	45.33	42.98	46.14	45.88	43.58	46.16	43.40	43.98	44.07	46.28	38.93
Distance activities	42.09	48.24	41.50	44.52	43.53	45.00	53.06	41.96	50.63	42.15	45.85	38.52	43.92	44.19
Social functioning	42.73	46.81	40.20	44.79	43.69	44.64	42.38	44.37	46.79	43.22	43.24	46.25	45.52	40.63
Mental health	46.38	38.72	31.47 *	46.61 *	42.14	47.91	42.25	44.39	48.03	42.88	44.51	42.50	43.46	45.20
Role difficulties	46.76	37.87	50.73	42.60	43.85	44.32	33.22	46.43	41.45	44.71	41.75	50.64	42.57	47.19
Dependency	47.78 *	35.59 *	47.40	43.29	43.80	44.43	30.03 **	47.15 **	40.00	45.12	40.42 *	54.59 *	43.23	45.70
Driving	48.08 *	34.93 *	49.07	42.94	44.81	42.30	47.06	45.56	45.11	43.69	42.25	49.18	42.85	46.56
Color vision	48.08 ***	34.93 ***	46.53	43.47	42.26	47.66	34.00 *	46.25 *	42.74	44.35	42.91	47.23	42.61	44.62
Peripheral vision	45.75	40.11	42.40	44.33	44.13	43.73	34.44 **	46.15 *	39.74	45.19	43.33	45.98	43.67	44.72
Total score	47.73 *	35.72 *	39.33	44.97	44.33	43.30	29.97 **	47.16 *	47.82	42.93	42.52	48.36	44.99	41.80

NEI-VFQ-25: National Eye Institute Visual Functioning Questionnaire-25; M: male; F: female; AHT: arterial hypertension. * *p* ≤ 0.05, ** *p* ≤ 0.01, *** *p* ≤ 0.001.

**Table 3 healthcare-10-00142-t003:** Correlation between the subscales and the total score of the NEI-VFQ-25 and continuous variables.

NEI-VFQ-25	Age	Diabetes Duration	Retinopathy Duration	HbA1c	BCVA
General health	−0.132	−0.057	0.102	0.018	−0.028
General vision	0.189	−0.113	−0.151	−0.064	−0.364 **
Ocular pain	−0.133	−0.090	0.116	−0.186	−0.127
Near activities	0.106	−0.077	0.040	0.022	−0.386 **
Distance activities	0.148	−0.018	−0.008	0.057	−0.396 **
Social functioning	0.131	0.033	−0.116	−0.040	−0.069
Mental health	0.103	−0.025	0.158	−0.008	−0.268 *
Role difficulties	−0.129	0.123	−0.012	0.021	0.499 **
Dependency	−0.119	0.078	−0.023	0.145	0.412 **
Driving	−0.040	0.196	0.065	0.071	0.435 **
Color vision	−0.103	0.083	−0.023	−0.066	0.385 **
Peripheral vision	−0.076	−0.044	−0.022	0.175	0.193
Composite score	0.004	0.039	−0.017	0.016	0.048

NEI-VFQ-25: National Eye Institute Visual Functioning Questionnaire-25; HbA1c: glycosylated hemoglobin A1c; BCVA: best-corrected visual acuity. * The correlation is significant at the 0.05 level (bilateral). ** The correlation is significant at the 0.01 level (bilateral).

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
