# Peer review of "Exploring the Quality of Life Related to Health and Vision in a Group of Patients with Diabetic Retinopathy"

_healthcare, 2022, doi:10.3390/healthcare10010142_

Round 1

Reviewer 1 Report

Dear Editor, Dear Authors,

Thank you for giving me the oppurtunity to read this work. 

I would like to share with you the following comments that could be helpful in improving this work:

Introduction:

Line 32. “among the urban population of developing countries”; This statement concerns developed or developing countries? The subject matter is also of importance in developed countries, like Spain.

Please, explain differences between general quality of life and disease-related quality of life.  Line 61 to 73 explains the effect of DR on quality of life but does not inform the reader if this literature review is related to general quality of life or disease related quality of life (DR related quality of life), especially that the study chose a specific quality of life measure. A justification for the choice of the measure, that is specific to decreased vision, is warranted. It is indeed a good choice but, in my opinion, it needs to be justified/explained.

The objective of the study is not clearly stated. It suggests a description of the QoL while associations and correlations are carried out. I suggest a reformulation of the main objective of the study and a limitation of the number of variables studied, given a sample size of only 87 participants.

Please define what is “episodes of decompensated diabetes”.

“Subjects with an incomplete medical history were excluded”; what are elements or items of the medical history? - explained at a later stage

Line 83 “which had an appointment with specialized in DR”; who is specialized in DR? is it a medical doctor? A diabetologist?

The inclusion criteria, other than age and a diagnosis of DR, are not described

Who collected the data?

Line 91 to 95: not sure that this is the right place to discuss study interruption.

Tables are not well presented.

I don’t think there is a need to repeat the p values in the narratives if they are already mentioned in the tables.

The sample size seems small to account for all the variables (number of variables and categories in categorical variables) and all the statistical tests carried out.

Reviewer 2 Report

The manuscript studies the quality of life related to health and vision in a group of patients with DR.

The results are shown in tables. It would be better to add some graphs or figures to show the results.

Reviewer 3 Report

Dear Authors,

This is interesting paper concerned important problem of quality-of-life patients with diabetes complications. But I want to underline a few problems which I saw:

  1. Abstract has no information about the aim of the study, results, and real conclusion.
  2. Lines 39-41- poorly controlled diabetes has hyperglycemia- it is doubled in one sentence- microvascular complications are consequences of poorly controlled diabetes.
  3. In whole manuscript- should be used: type 1 diabetes and type 2 diabetes with abbreviations T1D and T2D, only if we write about diabetes at all we could use term diabetes.
  4. Lines 61-64- doubled sentences- one should be deleted
  5. Lines 93-95- this sentence is completely not necessary.
  6. Main Outcomes- Instruments- Questionnaire used in the study should be much more precisely presented, maybe even as attachment. Methodology of the study should be reproductible, and precisely described. There is lack of references to the source of this questionnaire. No information about counts for each category.
  7. Line 115 and whole manuscript (lines 143, 157, 231 table 1, table 2- it has to be changed- it could not be used in manuscripts- insulin-dependence-but insulin treatment, insulin requirement, but not dependence- insulin dependent is only type 1 diabetes. Any of the T2D are not dependent on insulin.
  8. Table 1- there is lack of information of patients age and diabetes duration, and retinopathy duration, but later there is analysis of this parameters.
  9. Table 1 – episodes of diabetic decompensations requiring hospital admission- it was before study, or in time of study? it was the reason for hospital admission and then inclusion to the study? - it is not clear for reader

Author Response

Response to Reviewer 3:

First of all, we would like to express our sincere gratitude for all comments and suggestions received from the Reviewer 3. This information has certainly enriched the text for its best understanding, thank you very much indeed. We have clarified the reviewer3’s questions. We have introduced the required changes both in our answers to the specific comments and in the final manuscript V2. We have also made language-related changes throughout the manuscript.

Broad comments:

Abstract has no information about the aim of the study, results, and real conclusion.

Response: Thank you very much for pointing it out. We have changed the abstract:

Abstract: (1) Background: Visual impairment of people with diabetic retinopathy (DR) and its high impact on the different dimensions of their lives can cause a significant deterioration in the quality of life. The aim of this study was to study the association and relationship between quality of life related to vision and relevant clinical and sociodemographic variables in a group of patients with DR in Spain. (2) Methods: A descriptive cross-sectional study was conducted in all patients with DR over 18 years under follow-up in the Retina Service of the University Hospital of Burgos (HUBU), recruited during the months of January and February 2020. The main study variable was quality of life related to health and vision, obtained using the National Eye Institute Visual Function Questionnaire 25 (NEI-VFQ-25). (3) Results: 87 participants made up the sample, and significant differences were found in the NEI-VFQ-25 according to gender, type of diabetes, episodes of de-compensated diabetes and high blood pressure (HBP) (p <0.05). Best corrected visual acuity (BCVA) was also correlated with the NEI-VFQ-25 (p <0.05). (4) Conclusions: These data could facilitate the design of action protocols focused on the well-being of the patient, in addition to considering the clinical characteristics. Further studies are needed to help to understand the causal relationship between variables and that include a wider variety of factors.

Lines 39-41- poorly controlled diabetes has hyperglycemia- it is doubled in one sentence- microvascular complications are consequences of poorly controlled diabetes.

Response: Thank you very much for pointing it out. We have made the following changes:

“Diabetic retinopathy (DR) is one of the microvascular complications of long-term poorly controlled DM [6].”

In whole manuscript- should be used: type 1 diabetes and type 2 diabetes with abbreviations T1D and T2D, only if we write about diabetes at all we could use term diabetes.

Response: Thank you very much for pointing it out. We have change the required changes.

Lines 61-64- doubled sentences- one should be deleted

Response: Thank you very much for pointing it out. We have deleted the last sentence.

Lines 93-95- this sentence is completely not necessary.

Response: Thank you very much for pointing it out. We have deleted this sentence.

Main Outcomes- Instruments- Questionnaire used in the study should be much more precisely presented, maybe even as attachment. Methodology of the study should be reproductible, and precisely described. There is lack of references to the source of this questionnaire. No information about counts for each category.

Response: Thank you very much for pointing it out. We have added information to the Main Outcomes-Instruments section:

“The main study variable was quality of life related to health and vision. It was ob-tained using the NEI-VFQ-25 [26], a self-administered visión-specific patient-reported outcome outcome measure, reporting on visual function in everyday life, that has been validated worldwide across different ocular diseases. NEI-VFQ 25 has also been considered as a tool that is sensitive to change in visual acuity as it measures general health, general vision, eye pain, difficulty in near vision, difficulty in distant vision activities, limitation in social functions due to vision, mental problems due to vision, role problems due to vision, de-pendence on other people, driving difficulties, colour vision problems and peripheral vi-sion difficulties. The total score can range from 0 to 100; and higher scores indicate better results [26]…”

Line 115 and whole manuscript (lines 143, 157, 231 table 1, table 2- it has to be changed- it could not be used in manuscripts- insulin-dependence-but insulin treatment, insulin requirement, but not dependence- insulin dependent is only type 1 diabetes. Any of the T2D are not dependent on insulin.

Response: Thank you very much for pointing it out. We have added the required changes in the whole manuscript.

Table 1 – episodes of diabetic decompensations requiring hospital admission- it was before study, or in time of study? it was the reason for hospital admission and then inclusion to the study? - it is not clear for reader

Response: Thank you very much for pointing it out. We have explained "episodes of diabetic decompensations" in more detail in the Main Outcomes-Instruments section.

Paula Rodríguez Fernández